# HABIT: Health visitors delivering Advice in Britain on Infant Toothbrushing – an early-phase feasibility study of a complex oral health intervention

Erin Giles ![ORCID],[1] Faye Wray,[2] Ieva Eskyte ![ORCID],[3] Kara A Gray-Burrows ![ORCID],[1] Jenny Owen,[1] Amrit Bhatti,[1] Tim Zoltie ![ORCID],[1] Rosemary McEachan ![ORCID],[4] Z Marshman ![ORCID],[5] Sue Pavitt ![ORCID],[1] Robert M West ![ORCID],[2] Peter F Day ![ORCID] [1,6]

[1]School of Dentistry, University of Leeds, Leeds, UK
[2]Leeds Institute of Health Sciences, University of Leeds, Leeds, UK
[3]School of Law, University of Leeds, Leeds, UK
[4]Born in Bradford, Bradford Institute of Health Research, Bradford, UK
[5]University of Sheffield Faculty of Medicine Dentistry and Health, Sheffield, UK
[6]Community Dental Service, Bradford District Care NHS Foundation Trust, Bradford, UK

**Correspondence to**
Erin Giles; e.giles@leeds.ac.uk

## ABSTRACT

**Objectives** To conduct an early-phase feasibility study of an oral health intervention, Health visitors delivering Advice on Britain on Infant Toothbrushing (HABIT), delivered by Health Visitors to parents of children aged 9–12 months old.

**Design** A mixed-methods, early-phase, non-controlled, feasibility study.

**Participants** Recruitment consisted of Group A—HABIT-trained Health Visitors (n=11) and Group B—parents of children aged 9–12 months old about to receive their universal health check (n=35).

**Setting** Bradford, West Yorkshire, UK.

**Intervention** A multidisciplinary team co-developed digital and paper-based training resources with health visitors and parents of young children. The intervention comprised of two components: (A) training for health visitors to deliver the HABIT intervention and (B) HABIT resources for parents, including a website, videos, toothbrushing demonstration and a paper-based leaflet with an oral health action plan.

**Primary and secondary outcome measures** Recruitment, retention and intervention delivery were analysed as key process outcomes for Groups A and B. Group B demographics, self-reported toothbrushing behaviours, dietary habits and three objective measures of toothbrushing including plaque scores were collected at baseline, 2 weeks and 3 months post intervention.

**Results** HABIT intervention delivery was feasible. Although the intended sample size was recruited (Group A=11 and Group B=35) it was more challenging than anticipated. Retention of Group B participants to final data collection was satisfactory (n=26). Total compliance with toothbrushing guidelines at baseline was low (30%), but significantly improved and was maintained 3 months after the intervention (68%). Plaque scores improved post intervention and participants found video recording of toothbrushing acceptable. Dietary habits remained largely unchanged.

**Conclusion** This feasibility study has demonstrated that HABIT is an appropriate oral health intervention. Adaptions

## STRENGTHS AND LIMITATIONS OF THIS STUDY

⇒ Health visitors delivering Advice in Britain on Infant Toothbrushing (HABIT)—a complex, co-designed oral health behavioural intervention—has been developed using Medical Research Council guidance.

⇒ This early-phase feasibility study analysed recruitment, retention and intervention delivery as key process outcomes.

⇒ Data were collected in the home setting and included clinical, observed and self-reported toothbrushing and dietary habit data collected at three time points: baseline and at 2 weeks and 3 months post HABIT intervention.

⇒ Self-reporting of toothbrushing and dietary habits has limitations, which the authors recognise.

⇒ Challenges relating to recruitment and retention of participants will need to be addressed before progression to definitive study or full trial.

to the study design are recommended to maximise recruitment and data collection in a definitive study. These quantitative findings have demonstrated an early signal of impact for improved oral health behaviours for young children at high risk of decay.

**Trial registration number** ISRCTN55332414.

## BACKGROUND

Untreated dental caries is the most common health condition in the world. It is currently estimated that over half a billion children worldwide suffer from caries in their primary teeth.[1] In England, a quarter of children have experienced caries by the age of 5.[2] However, wide disparities exist. Caries rates in deprived areas are around double the national average.[2] The sequelae of untreated dental disease has significant consequences for the child, their family and wider society—negative consequences include pain, infection,

time off school, time off work for parents and a poorer quality of life.[3–5]

Caries is a pressing public health priority owing to its prevalence and impact. Critically, however, it is preventable. Optimal home-based oral health habits established in early childhood provide long-term protective benefits.[6] National guidelines in England,[7] provide clear, evidence-based definitions for these 'optimal' home-based oral health practices, including; brushing a child's teeth twice a day with a smear of at least 1000 ppm fluoridated toothpaste as soon as the first tooth erupts into the mouth, and limiting the frequency and amount of sugary foods and drinks.[7]

Health visiting teams in England provide a crucial role in giving children the best possible start in early childhood by signposting families to support, encouraging positive health behaviours and safeguarding children at risk.[8] Five mandatory universal visits are undertaken by the health visiting team during the first 2 years of each child's life, the fourth visit takes place when the child is aged 9–12 months old.[9] During this visit topics such as child development, health, nutrition and obesity prevention are discussed. These visits can be undertaken by health visitors or appropriately trained nursery nurses. In this paper the term 'Health Visitors' will be used collectively to represent any health visiting team member who took part in the HABIT intervention.

The first 1000 days of a child's life are generally accepted to be the most important in their physical, psychological and emotional development.[10 11] As part of the Healthy Child Programme,[12] oral health promotion is a key topic to be covered during these mandatory visits. The eruption of an infant's first tooth occurs at around 6 months of age,[13] therefore the fourth universal health visit is usually the first opportunity for toothbrushing guidance to be discussed. Health Visitors, however, have identified barriers in providing oral health advice including a lack of specialist knowledge, training, resources, navigating difficult conversations and recognising oral health as a priority.[14–18]

'Health visitors delivering Advice in Britain on Infant Toothbrushing' (HABIT) is a co-designed complex oral health intervention, underpinned by behaviour change theory.[19 20] Feasibility studies are a critical step in the intervention development process; exploring acceptability, feasibility of delivery, recruitment and retention of participants and anticipating problems prior to a large-scale randomised control trial (RCT).[21]

### Aims and objectives
The primary aim was to:
► Conduct an early-phase feasibility study of an oral health intervention, HABIT, delivered by Health Visitors to parents of children aged 9–12 months old.
The objectives were to:
► Analyse recruitment and retention of participants of both Health Visitors and parents.

► Assess the feasibility of intervention delivery and data collection.
► Explore the potential impact of HABIT on optimal oral health behaviours.

### METHOD
#### Study design
This was a mixed-methods, early-phase, non-controlled, feasibility study. Health Visitors and parents of children aged 9–12 months were recruited, all receiving the intervention.

#### Patient and public involvement
Participants were involved in the design, conduct, reporting and dissemination of this research project. Using a series of focus groups, previously designed health-visiting oral health resources were discussed and evaluated by Health Visitors and parents of young children. These discussions, in combination with the Medical Research Council framework,[22] informed the development of the intervention. A designated member of the research team supported and kept in regular contact with participants, allowing real-time feedback and alterations to the study conduct. Intervention and study design was acceptable to participants and is reported in a separate paper exploring the qualitative findings.[23] In addition, key stakeholders were invited to a research dissemination event held in Bradford in which preliminary findings were presented and feedback sought. Participation included representatives from: Health Visiting and dental teams, Bradford District Care NHS Foundation Trust, Bradford Metropolitan District Council, the British Society of Paediatric Dentistry, Oral Health Strategy Group and Public Health England.

#### Intervention outline
HABIT is an oral health intervention, underpinned by complex behavioural change theory, undertaken in a home setting. It comprises of two parts: (1) Training Health Visitors to deliver the intervention; (2) Delivery of the HABIT intervention by trained Health Visitors at the 9–12 month universal health visit. Oral health conversations are supported by HABIT resources for parents, including a website, videos, toothbrushing demonstration and a paper-based leaflet with an oral health action plan. A detailed summary of the HABIT intervention is outlined using the Template for Intervention Description and Replication (TIDieR) checklist[24] in online supplemental item 1.

#### Setting
Bradford is a metropolitan borough located in the UK, situated within the northern county of West Yorkshire. It is the fifth largest city in the UK with a population of approximately half a million. Bradford's urban areas are among the most deprived in the UK.[25] The population of Bradford is also ethnically diverse—64% of the population identify as white British, with 20% of the population of Pakistani ethnic origin.[26]

## Recruitment and retention

Group A participants were Health Visitors who provide the 9–12 month universal health check. Following grant funding, the research team worked with key contacts from the local health visiting team to disseminate research information. All eligible health visitors were invited to take part via email and those that volunteered to participate attended a 1-day training course provided by the researchers.

Group B participants were recruited from the Bradford waiting list of children due to have their 9–12 month universal check. For each HABIT-trained Health Visitor, a report of their next 20 visits was identified. An invitation letter, project information sheet and consent form were posted with the standard universal check reminder letter. For Group B:

### Inclusion criteria
► Parents of a child aged 9–12 months old who were about to receive a universal home visit by a HABIT-trained Health Visitor.
► Child with at least one erupted tooth.

### Exclusion criteria
► Non-English-speaking families, as interpreter services were not available to the research team.

An NHS Trust Clinical Studies Officer contacted parents by phone to explore if they wanted to participate. If the invitation was accepted, they would visit the parent to gain written consent and organise a home visit for baseline data collection. As suggested in good practice recommendations,[27] we intended to recruit 30 parents–children (dyads) to our feasibility study. Thirty participants would provide a 95% CI for prevalence of no wider than ±19.6%, anticipating a minimum of 15% loss to follow-up.

## Data collection

Data collection was carried out by three dental professionals experienced in providing care for children (SH, JO and HG), supported by three research assistants (KT, IE and FW). An experienced British Association of the Study of Community Dentistry (BASCD) examiner provided training to ensure a consistent approach to inspection procedures, BASCD caries assessment criteria and tooth codes.[28] Dental researchers underwent calibration for caries and plaque detection, with agreement calculated using Fleiss' kappa.

Data were collected in a home setting. Children were examined supine, using a disposable dental mirror and a head torch for illumination. Cotton rolls were available, if required, to remove debris. Between the baseline and second data collection visit, parents received the HABIT intervention delivered by a Health Visitor as part of the child's universal 9–12 month health visit. Further data collection was carried out 2 weeks and 3 months following the intervention. A flowchart of study design is outlined in Figure 1.

1. First visit—baseline (BL).

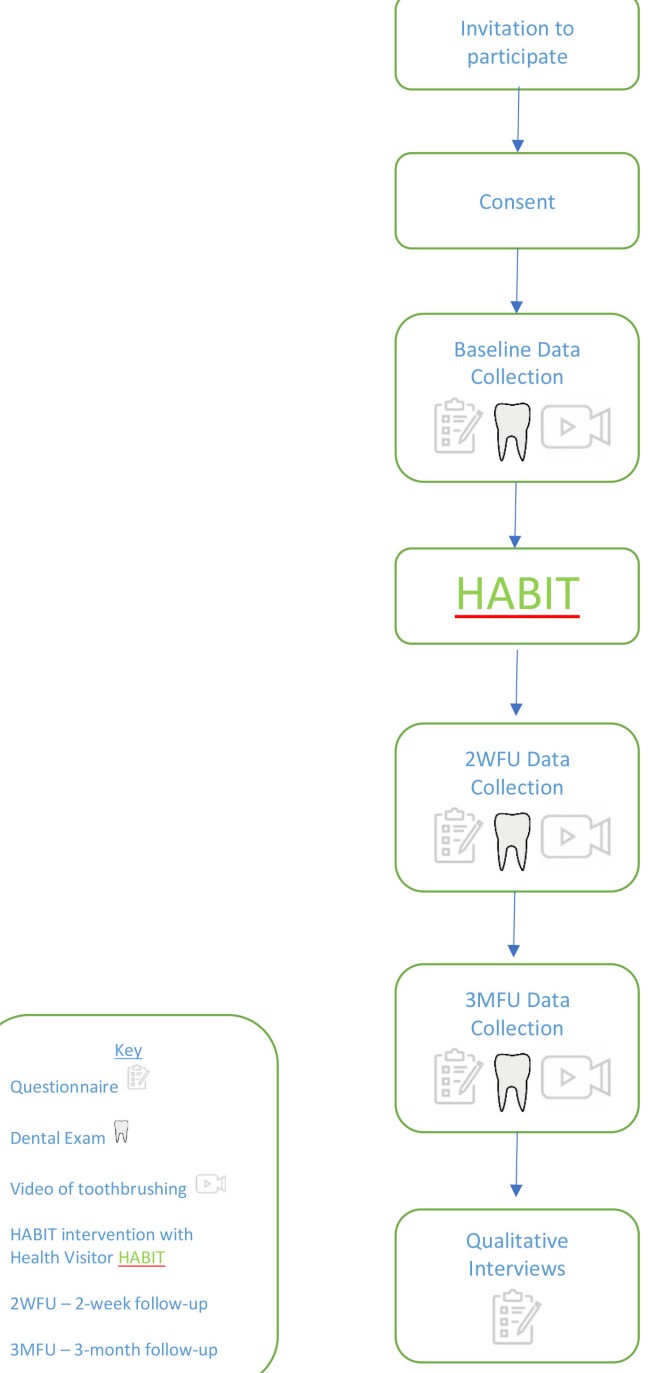

**Figure 1** Study design. HABIT, Health visitors delivering Advice in Britain on Infant Toothbrushing; 3MFU, 3-month follow-up; 2WFU, 2-week follow-up.

2. Self-reported toothbrushing and dietary behaviours were collected through a questionnaire. Toothbrushing diaries were issued as an additional, contemporaneous, measure of toothbrushing frequency. Three objective measures of parental supervised brushing (PSB) were collected:
   a. Teeth were examined for cavitated dentinal caries and restorations using the BASCD criteria.[28]
   b. Children's pre-brushing plaque levels per sextant using the Oral Hygiene Index.[29]

c. Duration of toothbrushing: researchers filmed parent–child toothbrushing for subsequent evaluation.[30]

3. Second visit—2-week follow-up (2WFU)

Two weeks following the HABIT intervention, further self-report questionnaire data and objective measures of PSB (a–c above) were collected at a home visit.

4. Third visit—3-month follow-up (3MFU).

Three months following the HABIT intervention, further self-report questionnaire data and objective measures of PSB (a–c above) were collected at a home visit. A £10 gift voucher was issued after each visit.

### Self-reported oral health behaviours

The validated questionnaire[31 32] collected information on participant socio-demographic data, self-reported toothbrushing, dietary behaviours and feeding practices. Self-report determinants of toothbrushing were measured against national guidance[7] outlining five key items for toothbrushing: parental supervision, 1000 ppm toothpaste fluoride concentration, smear of toothpaste, twice-daily toothbrushing and wiping away excess toothpaste at the end of brushing. A compound measure of 'total' compliance was also calculated, identifying those fully compliant to all five items. Contemporaneous data on the frequency of toothbrushing was collected by a paper diary. Dietary data were collected using an established questionnaire trialled within a Bradford birth cohort of similar aged children,[33] measuring frequency of consumption of the main food groups. Frequency scoring was non-linear (eg, 0=none, 1=less than once a month, 2= one to three times a month, etc.) and covered all aspects of diet with a focus on sugary snacks, sweetened beverages and fruit and vegetable intake.

### Dental examination objective measures of toothbrushing

Teeth were examined for cavitated caries and dental restorations using the BASCD criteria.[28] A score was calculated based on the number of decayed, missing and filled teeth (dmft). Prior to performing the Oral Hygiene Index, allergy status was confirmed (checking no allergy to food colourings) and petroleum jelly was applied to the lips. The infants' teeth were dyed using TePe PlaqSearch disclosing solution applied with a small dental brush. Once the plaque index (score) had been taken, gauze was used to wipe the dyed plaque from the teeth to ensure it did not act as a visual aid for the parent in the subsequent videotaping of toothbrushing. Partially completed plaque scores were discounted in the statistical analysis.

### Videotaping of child

Videotaping of child–parent toothbrushing was undertaken using a small action camera (GoPro HERO 5, Go Pro). Practicability of video recordings pertains to the feasibility of data collection and was therefore analysed alongside toothbrushing duration. The videos also provided an objective method of assessment with national toothbrushing guidelines[7] and will be reported in a separate paper.

### Data analysis

Data relating to recruitment, retention, intervention delivery and feasibility of data collection were analysed using descriptive statistics in MS Excel. Progression criteria to definitive study were agreed: >25% recruitment and >75% retention—automatic progression; 20–25% recruitment and 65–74% retention—modifications recommended; <20% recruitment and <64% retention—consultation with advisory group and remedial actions required.

To explore potential oral health impacts, CIs for continuous variables (eg, plaque scores) were calculated using a t-distribution. For categorical outcomes, the CIs for rate were calculated based on the binomial distribution. In respect to self-reported toothbrushing compliance, repeated measurements were fitted to a multilevel model with time points nested within dyads and allocating a random intercept for dyad. Maximum likelihood fitting was used rather than reduced maximum likelihood to permit hypothesis testing. Within the model, a term was added for 'time points' increasing the df by 2. This enabled the reporting of compliance rates at each time point and formal statistical testing of the effect of the intervention over time.

The statistical significance of the time term was then determined by use of the log-likelihood ratio test. Similarly, a two-level linear regression was fitted for plaque scores and changes over time tested through a log-likelihood test. This analysis was valid under the assumption that drop out was at random. The analysis was undertaken in the R statistical software environment[34] and using the lme4 package.[35]

## RESULTS

### Recruitment and retention

Eleven Health Visitors were recruited to Group A (n=11) to deliver the HABIT intervention. Of the 11, eight (73%) including six health visitors and two nursery nurses, had the opportunity to deliver the intervention, as outlined in Figure 2.

Group B dyads were recruited over a 10-month period between December 2017 and October 2018. From 127 invitations, 35 dyads were recruited and consented—a recruitment rate of 28%. Anticipating a minimum of 15% loss to follow-up, the achieved sample size of n=35 was sufficient. Nine participants were lost, resulting in a retention rate of 74% (n=26). Almost all dropouts occurred between consent (n=8) and prior to BL data collection. Figure 2 shows the participant flow and reasons for dropout for Group B participants.

### Demographics

The BL demographic data is shown in Table 1. There was significant variation in education, employment status and

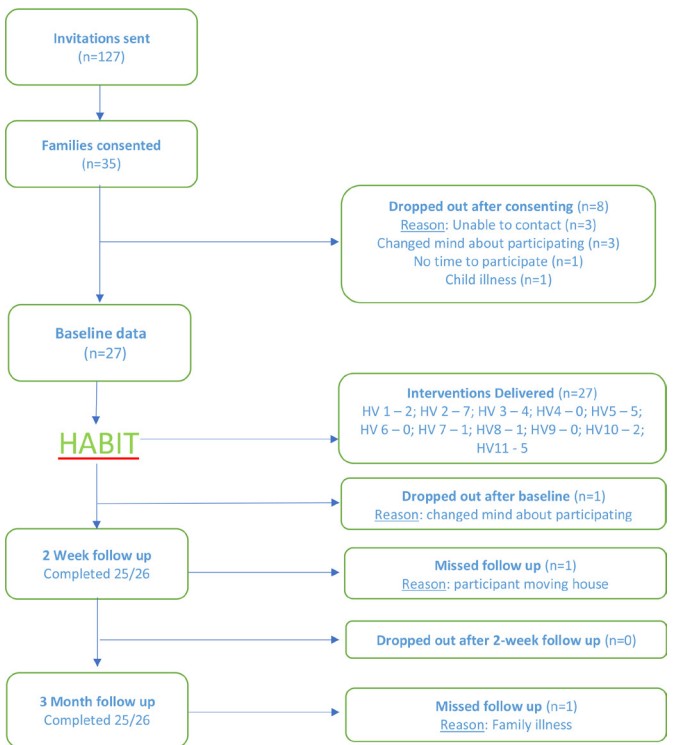

**Figure 2** Participant flowchart, as outlined by Consolidated Standards of Reporting Trials.[35]

**HABIT: Health visitors delivering Advice in Britain on Infant Toothbrushing**

**HV: Health Visitor**

family group types. Almost half (n=12) of the dyads lived in the most deprived decile area of England, according to the Index of Multiple Deprivation (IMD) 2019.[25] The average age of a parent in the study was 31, and the majority (n=21) were born in the UK. No child in the study had caries, an unsurprising finding given the age of the children.

### Feasibility of data collection and intervention delivery

Out of a possible 79 home visits, the research team were able to undertake 97% (n=77) of planned data collection. Reasons for the two missed visits are outlined in Figure 2. Of the 27 dyads who completed BL data collection, all received the HABIT intervention.

### Intervention outcomes
#### Self-reported toothbrushing behaviours
For self-reported toothbrushing habits, 'total' compliance improved from 30% (n=27; 95% CI 0.13 to 0.47) at BL to 70% (n=24; 95% CI 0.53 to 0.89) 2 weeks and 68% (n=25; 95% CI 0.50 to 0.86) 3 months following the HABIT intervention. These findings were statistically significant and are shown in Table 2. After the intervention, all parents had commenced toothbrushing. Compliance to all individual components of the guideline increased, although insignificantly, after the intervention and are outlined

| Table 1 | Baseline demographic data |
|---|---|
| **Characteristic** | **N (%)** |
| Age of parent | |
| Mean | 31 [S.D: 6.0] |
| Range | 19–42 |
| % over 30 | 19 (70) |
| Birthplace | |
| UK | 21 (78) |
| Pakistan | 3 (11) |
| Other | 3 (11) |
| Number of children | |
| Multichild house | 17 (63) |
| Qualifications | |
| Less than five GCSEs* | 8 (30) |
| A-level or equivalent† | 10 (37) |
| University degree | 9 (33) |
| Employment | |
| Currently employed | 13 (48) |
| Household income | |
| Less than £16 100 | 3 (11) |
| £16 100–£21 249 | 7 (26) |
| £21 250–£27 999 | 1 (4) |
| £28 000–£38 399 | 2 (7) |
| £38 399+ | 5 (19) |
| No answer/unsure | 9 (33) |
| Finances | |
| 'Living Comfortably' | 11 (41) |
| 'Doing Alright' | 8 (29) |
| 'Just Getting By' | 1 (4) |
| 'Finding It Difficult' | 0 (0) |
| No answer/unsure | 7 (26) |
| IMD centile | |
| 1st most deprived | 12 (44) |
| 2nd | 2 (7) |
| 3rd | 4 (15) |
| 4th | 4 (15) |
| 5th | 2 (7) |
| 6th | 2 (7) |
| 7th | 1 (4) |
| 8th | 0 (0) |
| 9th | 0 (0) |
| 10th least deprived | 0 (0) |

*GCSE: General Certificate of Secondary Education
†A-level: Advanced Level Qualification
IMD, Index of Multiple Deprivation .

in Table 2. There was poor uptake of the toothbrushing diaries that generated little usable data.

### Dietary habits
Online supplemental item 2 illustrates the median consumption of notable food and drink categories over the course of the study, along with the frequency key

**Table 2** Self-report measures of toothbrushing pre-intervention and post-intervention with effect estimate (95% CI)

| Compliance to 'Delivering Better Oral Health' (DBOH) guidelines | Baseline n=27 % (95%CI) | Two-week follow-up n=24* % (95% CI) | Three-month follow-up n=25† % (95% CI) |
|---|---|---|---|
| Using ≥1000 ppm fluoride toothpaste | 81 (0.66 - 0.96) | 100 (1.00 - 1.00) | 100 (1.00 - 1.00) |
| Using a smear of toothpaste | 63 (0.45 - 0.81) | 88 (0.75 - 1.00) | 88 (0.75 - 1.00) |
| Parental brushing | 81 (0.66 - 0.96) | 100 (1.00 - 1.00) | 100 (1.00 - 1.00) |
| No rinsing after brushing | 59 (0.40 - 0.78) | 92 (0.88 - 1.00) | 88 (0.75 - 1.00) |
| Brushing at least twice a day, including at night | 56 (0.37 - 0.75) | 88 (0.75 - 1.00) | 88 (0.75 - 1.00) |
| **Total compliance to all DBOH guidelines** | 30 (0.13 - 0.47) | 71 (0.53 - 0.89) | 68 (0.50 - 0.86) |

*Two self-reported brushing data sets not available.
†One self-reported brushing data set not available.

used for coding. The consumption of biscuits decreased (BL: 4 and 3MFU: 2) and sweets increased (BL: 0 and 3MFU: 1), however, there were no significant changes in infant diet after the intervention. Use of free flow cups increased from 41% (n=11) at BL to 72% (n=18) at the final follow-up visit.

## Plaque scores
The research team fully completed 67 (85%) plaque scores. The most common reasons for incomplete plaque scores were behavioural challenges or the infant being asleep. There was an incremental decrease in plaque scores between BL (42%, n=25; 95% CI 0.23 to 0.61), 2WFU (20%, n=21; 95% CI 0.03 to 0.37) and the 3MFU (19%, n=21; 95% CI 0.02 to 0.36). The statistically significant decrease in plaque scores is represented in Figure 3.

## Videos of toothbrushing
From 79 data collection opportunities, video recording of toothbrushing was available for 75% (n=59) of home

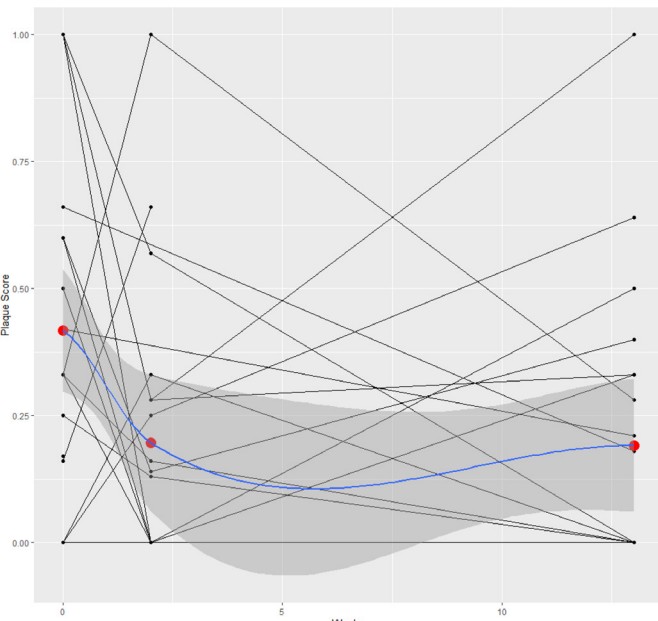

**Figure 3** Plaque score trends between baseline, 2-week follow-up and 3-month follow-up data collection visits.

visits. Reasons for no video recording were as follows: five (n=5) dyads were not brushing at BL; five babies were asleep at the time of their home visit (n=5); three parents could not find their child's toothbrush at the BL visit (n=3), one parent declined to consent for videotaping (n=1), two missed visits (see Figure 2) and two technical issues with the video camera (n=2). Toothbrushing duration increased from an average of 36 seconds at BL (n=18, SD=23.9) to 47 seconds at the final follow-up visit (n=19, SD=23.6).

## Inter examiner reliability
Inter examiner reliability of scoring for dental examinations and plaque scores was assessed with Fleiss' kappa, where a value of 0.85 showed that scores were significantly different from results that would have been obtained at random (p<0.001).

## DISCUSSION
### Recruitment and retention
Adequate recruitment and retention of participants are key outcome measures of any feasibility study.[21] Progression criteria should ideally be outlined in the protocol[20]; however, recommendations were first suggested in 2016 and this implementation has taken time.[36] Having achieved 28% recruitment and 74% retention of Group B, it is reasonable to recommend a conditional progression.

The intended recruitment and retention of Group A was achieved and feasible, however, the research team encountered some challenges. Initially, six Health Visitors were recruited to Group A but after three staff members went on long-term sick leave, a further five were recruited to allow intervention delivery to continue without delay. Since the transfer of health visiting to local authorities, councils have seen a £700 million cut to public health spending.[37] In the 3 years leading up to HABIT commencement, Health Visitor numbers dropped by nearly a quarter, with a concurrent rise in stress-related sickness.[38] This study has identified, to recruit in a suitable timescale and alleviate pressure on an already stretched workforce, it will be necessary to

increase intended recruitment numbers and continue to include wider members of the Health Visiting team.

Caries predominantly affects lower-socioeconomic areas of England, therefore Bradford is an ideal location for the implementation of oral health research. The families in Group B varied in ethnicity, education and income, and demographics were representative of the local area's high level of deprivation. Critically, this feasibility study recruited a diverse cohort within communities with high levels of oral disease.[25] Nearly half of the participants lived in the 10% most deprived areas of England, according to the IMD.[25] The average age of a parent in the study was 31, which is older than the average age of a first-time parent in England (28.9 years),[39] but is an expected finding given that most children (n=17) in the study were not first-born.

Approximately 28% of invited parents (Group B) were recruited to BL, reaching the target sample size. Recruitment took 10 months, 4 months longer than expected. Initially, the recruitment strategy included Health Visitors phoning up parents following the mail out of information, but Health Visitors struggled with this additional activity over and above their clinical workload. Following feedback, a member of the NHS Trust Research and Development team took over all recruitment responsibilities. Group B had a retention rate of 74% (n=26) at the final data-collection visit, a comparable level to similarly structured feasibility papers.[40] The biggest dropout occurred after consent (n=8). Combining the BL data collection and consent visit would streamline future study design and minimise attrition.[40] Parents with young children lead busy and complex lives, which can lead to communication challenges and last-minute cancellations, consequently, generous timelines can provide the required flexibility for home-based data collection.

### Feasibility of HABIT delivery and data collection

Intervention delivery and data collection in the home setting were feasible. The intervention targeted home-based oral health behaviours, therefore collecting data in this environment provides greater insight into household practices and parent–child interactions. Telephoning parents and organising visits within a tight time-period posed challenges; however, reasons for missed visits were generally due to unavoidable factors, such as child sickness. The data collection method may have discouraged participation for some, but this did not hamper anticipated recruitment rates. Families received a £10 voucher for each home visit, which helped participant retention.

### Intervention outcomes
#### Self-reported toothbrushing behaviours

As with any feasibility study using a small sample size, any suggestion of impact needs to be considered with caution. Nonetheless, we can report encouraging signs of improvement in optimum oral health behaviours following the HABIT intervention. At the final data collection visit there was a significant improvement in toothbrushing behaviours as shown by 'total' compliance with guidance. There were small increases in compliance to individual items of the guideline, however, these were non-significant within themselves. Importantly, by the end of the study, all parents had commenced infant toothbrushing with a fluoride toothpaste.

'Total' compliance to optimal toothbrushing behaviours was 30% at BL. A key finding from this feasibility study is the low level of compliance when the five-point guideline criteria is used.[7] Previous studies have shown much higher levels of compliance[41–43]; however, these studies did not use the full guideline criteria nor home-based data collection methods. Paper toothbrushing diaries were rarely completed by parents, therefore self-reported toothbrushing frequency in the questionnaire was used in the total compliance measure. A recent incentivised study found that children, on average, only brushed their teeth five times a week, using data transmitted from a Bluetooth-enabled electric toothbrush,[44] providing a clear justification for innovative data collection methods. Other contemporary data-collection methods, such as SMS messages sent each evening by parents also show initial signs of potential; 53% of their sample reported toothbrushing every evening.[45] These methods in conjunction with videos of toothbrushing may provide more accurate measurement tools to assess optimal toothbrushing behaviours.

To date, there is a lack of published studies in the UK looking at the effectiveness of oral health interventions in young children.[46 47] A recent Australian study has found that both telephone and home contacts by an oral therapist were effective at reducing caries experience and cost-effective when compared with routine care, although caution should be exerted in relation to the generalisability to a UK population.[48] A previous RCT found no significant difference in caries experience between children visited by an oral-health trained health visitor and routine care, however, methodological limitations and interference between study groups restrict the validity of the findings.[49]

### Dietary habits

There was little reported change in dietary habits over the course of the feasibility study. This is a positive finding, given there is often an increase in the consumption of sugary drinks and snacks during infancy, due to the weaning process and the increase in consumption of all food groups.[50 51] The continued intake of water and milk, as opposed to a transition to sweetened drinks, is also encouraging. Moreover, the use of 'free-flow' cups increased and valve-cups decreased, another important and recommended practice for infant feeding[7]

### Plaque scores

Plaque scores showed an encouraging and incremental decrease between BL visit and follow-up visits, however, undertaking plaque scores posed some challenges in this younger cohort. There were behavioural

and cooperation challenges; four plaque scores were partially completed and not included in statistical analysis. A study using a similar cohort of patients found that sensitivity and specificity of plaque scores was poor, although this was undertaken without the use of a disclosing agent.[52]

### Videos of toothbrushing

Filming of toothbrushing was acceptable to nearly all participants (n=26), however eight video opportunities were missed owing to logistical factors, such as the infant being asleep. Active parental toothbrushing was short in duration (36 secs), but showed a positive increase following the HABIT intervention (47 secs). A separate pilot study of toddlers demonstrated similar findings; despite toothbrushing sessions lasting, on average, over 2 minutes, active toothbrushing occurred for less than half this time.[53] This suggests that routine parental supervision of toothbrushing falls far short of professional expectations.

### Limitations

Both Health Visitors and parent–child dyads were recruited through positive response to a research invitation. It could be assumed that they represent a more motivated and 'aware' subsection of the larger population, and thus create a potential selection bias. To improve this in any larger trial or study incentives, such as monetary gifts prepaid to participants, could be used to aid recruitment and retention in underserved groups.[54] Interpreting services were not available to the research team, which will have limited the inclusion of participants who were not English-speaking. Liaising and working with parents closely will be key to the success of any future study. Arranging visits at a time when the infant is most likely to cooperate, such as avoiding nap time, will maximise the utility of plaques scores and videotaping.

Collecting information on infant dietary habits posed several difficulties. The questionnaire, although based on a substantiated data collection method[33] had shortcomings; it relied on memory, parents were sometimes unsure of diet when looked after by family and friends, and anecdotally there was confusion around the definition of some food groups and dietary terminology (eg, no 'added' sugar compared with sugar-free). Amendment of this data collection method is required prior to definitive trial. Certain dietary behaviours can be more easily and accurately collected through a focus on high-risk behaviours, such as sugar consumption in the hour before bedtime.[55]

## CONCLUSION

This early-phase feasibility study has demonstrated that HABIT is a feasible oral health intervention for Health Visitors to deliver to the parents of children aged 9–12 months old. Data collection and intervention delivery in the home setting was achievable. Challenges relating to recruitment, collecting dietary data and retention of participants prior to BL data collection need further modification prior to progression to a definite study or trial. The quantitative findings have shown an improvement in optimal oral health behaviours, suggesting an early signal of impact in a deprived population at high risk of caries.

**Acknowledgements** We would like to thank parents and health visiting team members including Susanne Gill, who participated in the study as well as Susan Hanslip, Helen Gledhill, Jane Holt and Kerina Tull for their help with data collection. We would like to acknowledge our Health Visiting colleagues, Victoria Smith, who coauthored the HABIT qualitative paper. It was our intention that Victoria would contribute in a similar manner to the HABIT quantitative analysis, however, she was unavailable to contribute to the final drafting of the paper. The views expressed in this publication are those of the author(s) and not necessarily those of the National Institute for Health Research (NIHR), the National Health Service or the Department of Health and Social Care. Further support for this feasibility study was provided by DenTCRU (Dental Translational and Clinical Research Unit), part of the NIHR Leeds Clinical Research Facility.

**Contributors** PFD is the principal investigator, guarantor, led the development of the grant and writing of the trial protocol. EG led the analysis and writing of the manuscript with major contributions from RMW, PFD and KAG-B. EG, AB, FW, JO and IE undertook data collection, data cleaning and monitoring aspects of the protocol. All authors including RM, ZM, TZ and SP have had the opportunity to read, contribute and approve the manuscript.

**Funding** The study was funded by the Medical Research Council (MR/P017185/1). Four of the authors of this paper (PFD, RM, ZM, and KAG-B) are supported by the National Institute for Health Research (NIHR) Applied Research Collaborations Yorkshire and Humber (NIHR ARC YH) NIHR200166 www.arc-yh.nihr.ac.uk.

**Competing interests** None declared.

**Patient and public involvement** Patients and/or the public were involved in the design, or conduct, or reporting, or dissemination plans of this research. Refer to the Methods section for further details.

**Patient consent for publication** Not applicable.

**Ethics approval** Ethical approval was obtained by the Research Ethics Committee (17/YH/0301) (Yorkshire and The Humber - Leeds East Research Ethics Committee), Health Research Authority (IRAS ID 230315) and National Institute for Health Research Clinical Research Network Portfolio Adoption. All methods were carried out in accordance with relevant guidelines and regulations. Participants gave informed consent to participate in the study before taking part.

**Provenance and peer review** Not commissioned; externally peer reviewed.

**Data availability statement** Data are available upon reasonable request. The data sets generated and analysed during the current study are available upon request from Dr Peter Day, p.f.day@leeds.ac.uk. These include transcripts of qualitative interviews and spreadsheets with the quantitative data. The data is kept for 20 years. Videos of child–parent toothbrushing are not available owing to the ethical assurances made to participants at the time of consent.

**ORCID iDs**
Erin Giles http://orcid.org/0000-0001-8631-4809

Ieva Eskyte http://orcid.org/0000-0001-9486-0033
Kara A Gray-Burrows http://orcid.org/0000-0002-1550-5066
Tim Zoltie http://orcid.org/0000-0003-3411-341X
Rosemary McEachan http://orcid.org/0000-0003-1302-6675
Z Marshman http://orcid.org/0000-0003-0943-9637
Sue Pavitt http://orcid.org/0000-0001-7447-440X
Robert M West http://orcid.org/0000-0001-7305-3654
Peter F Day http://orcid.org/0000-0001-9711-9638

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
