## [Reviewer comments · BMJ Open]

ARTICLE DETAILS

TITLE (PROVISIONAL)	HABIT: Health visitors delivering Advice in Britain on Infant Toothbrushing – An early-phase feasibility study of a complex oral health intervention
AUTHORS	Giles, Erin; Wray, Faye; Eskyte, Ieva; Gray-Burrows, Kara; Owen, Jenny; Bhatti, Amrit; Zoltie, Tim; McEachan, Rosemary; Marshman, Z; Pavitt, Sue; West, Robert; Day, Peter

VERSION 1 – REVIEW

REVIEWER	Yuan, Siyang University of Dundee, Dental Health Services Research Unit, School of Dentistry
REVIEW RETURNED	23-Jan-2022

GENERAL COMMENTS	This is a well written feasibility study to deliver a home-based intervention to improve oral health related behaviours of parents of infants. It shows innovation by using videoed toothbrushing and diary to record toothbrushing. Please find my detailed comments in the attached documents.
--

REVIEWER	Bağ, İrem Kütahya Health Sciences University
REVIEW RETURNED	26-Feb-2022

GENERAL COMMENTS	I congratulate you for this study, which comprehensively evaluates parent education and health worker-family cooperation, which is an important criterion for oral health.
--

REVIEWER	Beckett, Deanna University of Otago, Oral Sciences
REVIEW RETURNED	17-May-2022

GENERAL COMMENTS	Thank you for the opportunity to review this manuscript. This is an interesting paper, and I would like to see it published. At the moment it is a little confusing, with the authors talking about aspects of the study that were then not reported on. Please see below for my comments: Abstract: Missing a background section? Needs to be more concise. For example, no need to say 'Following a complex intervention methodology - just concisely outline your intervention. No mention of group A methods or results in abstract? Methods:
--

	Patient and Public Involvement: If you are not going to report the qualitative findings, then they should not be mentioned in your methods. No methodology provided for group A intervention? Is this a qualitative part? If the authors are not going to present the findings of this aspect of the study, then it should not be mentioned or discussed in this paper. Setting: The methods should present the location, but the justification for the setting should be in the background, or the discussion. Recruitments and Retention: The authors mention group A participants, and group B participants, but no description of methodology for group A provided? Videotaping of child: Again, if you are not going to report the findings of this, then it should not be in this paper. Results: Demographics: Only demographics for group B provided in table 1? Table 1: In the data column, percentages should be in brackets after the N. Eg. 21 (78%). The column header should specify what is contained in the column e.g. n (%). The only exception is the age of parent row, where you already specify what is being presented in the row subheadings, and mean should also have SD in brackets afterwards. E.g mean (SD) for the row subheading, and 31 (SD?) in the data column. Table 2: Same comments as for table 1, need column descriptor N (%), and effect estimate and CI should be presented in a separate column, with the column headed accordingly. Discussion: I would like to see more discussion around how your studies findings compared to other studies. Videos of Toothbrushing: You did not present findings on the toothbrushing videos, therefore this should not be discussed in your paper. You mention that this will be in another paper, therefore remove it from this one. Although, including the qualitative and video data could really make for a great paper? But if you decide not to include, then remove all mention of both parts entirely from this one.
--	--

VERSION 1 – AUTHOR RESPONSE

Reviewer	Comment	Authors' Reply
----------	---------	----------------

1	1. P5, line 47, "...management of resistant parents..." Please be mindful of the wording. I understand that in Motivational Interviewing there's the term of 'rolling with resistance' which means dealing with the resistance for behaviour change in patients. Although parents might be 'resistant' for a reason, special consideration should be given to the background factors as the environment shapes the behaviours. Otherwise we may end up 'blame the victim' by ignoring the fundamental factors including their life circumstances. I would advise you change the wording/expression here.	Thank you for identifying this, we have changed the expression from 'management of resistant parents' to 'navigating difficult conversations'.
1	2. P8, Line 55 "A research officer contacted parents by phone to explore if they wanted to participate...". For ethical consideration, please check if researcher is able to directly contact potential participants (i.e. parents) or parents were contacted by phone/mail by HV.	Parents were contacted by a clinical studies officer, acting as a gatekeeper within the health visiting team. To clarify meaning, we have changed the sentence to: 'An NHS Trust Clinical Studies Officer contacted parents by phone...'
1	3. P9 (data collection), 1st paragraph, please state what caries assessment criteria you used in this study. I know that you reported elsewhere. But it's worth repeating this information in this article for a better readership.	The following sentence has been added to the initial data collection paragraph: "An experienced British Association of the Study of Community Dentistry (BASCD) examiner provided training to ensure a consistent approach to inspection procedures, BASCD caries assessment criteria and tooth codes"
1	4. P10, line 11-12, please check guidelines on the amount of fluoride toothpaste for under-3s, is that 'smear size' or 'pea size'?	Thank you for identifying this error, it has been changed to 'smear of toothpaste'.
1	5. P10, line 14, please indicate with details how the 'compound measure of 'total' compliance was calculated.'	To clarify meaning, the sentence has been amended to: "A compound measure of 'total' compliance was also calculated, identifying those fully compliant to all five items".
1	6. P10, line 19, is that frequency scoring for once a day or once a month? Here is confusing for reading and please explain explicitly.	To clarify meaning, the sentence has been changed to "Frequency scoring was non-linear (e.g. 0=none, 1=less than once a month... 8=four-five times a day, 9=six times a day)"
1	7. Discussion/limitation of the study: The parents are mostly from the most deprived area. Regarding dietary habits, I wonder if any of the parents understand the sugary level of the snacking/drinking. For instance, they may think food/drink package showing 'no added sugar' as healthy	Researchers felt there was some confusion around the use of this term and therefore have had added this to the discussion: '... confusion around the definition of some food groups and dietary terminology (e.g. no 'added' sugar compared to sugar-free)'.

	food/drinks. This can be regarded as a potential limitation, which could inform your future study.	
1	8. In addition, I understand that this is a feasibility study. You have reported the challenges of videoing, completion of toothbrushing diary and checking plaque scores in infants. How this could inform your large scale study. This could be addressed in the Discussion.	The following sentence has been added to the limitations section of the discussion: "Liaising and working with parents closely will be key to the success of any future study. Arranging visits at a time when the infant is most likely to cooperate, such as avoiding nap time, will maximise the utility of plaques scores and videotaping." The authors feel we have adequately addressed alternatives for monitoring toothbrushing frequency in the "Self-reported toothbrushing behaviours" section of the discussion.
3	1. Abstract: a) Missing a background section? b) Needs to be more concise. For example, no need to say 'Following a complex intervention methodology - just concisely outline your intervention. c) No mention of group A methods or results in abstract?	a) We thank the reviewer for the comment, however, as the abstract follows the journal's guidelines for submission and to keep to word count, we have not added a background section. b) We have removed the wording 'Following a complex intervention methodology' from the methodology. c) We have added Group A information to the methodology and results in the abstract
3	2. Patient and Public Involvement: If you are not going to report the qualitative findings, then they should not be mentioned in your methods	We thank the reviewer for the comment; however, the authors wish to highlight the PPI aspects of the qualitative study, reported in a separate, open-access paper. We have amended the sentence to highlight the key finding: "Intervention and study design was acceptable to participants and is reported in a separate paper exploring the qualitative findings."
3	3. No methodology provided for group A intervention? Is this a qualitative part? If the authors are not going to present the findings of this aspect of the study, then it should not be mentioned or discussed in this paper.	Although acceptability and process outcomes for Group A are explored in a separate paper, recruitment and retention of Group A is key to the exploration of feasibility in this early phase study.
3	4. Setting: The methods should present the location, but the justification for the	Justification for location has been moved to the discussion section of

	setting should be in the background, or the discussion.	the paper.
3	5. Recruitments and Retention: The authors mention group A participants, and group B participants, but no description of methodology for group A provided?	We have added the following information to the methodology: 'Following grant funding, the research team worked with key contacts from the local health visiting team to disseminate research information. All eligible health visitors were invited to take part via email and those that volunteered to participate attended a one-day training course provided by the researchers.'
3	6. Videotaping of child: Again, if you are not going to report the findings of this, then it should not be in this paper.	Brushing duration taken from the videotapes was analysed as part of the quantitative analysis, therefore, the authors feel it is necessary to include this section. Parent-child interaction video will be addressed in a separate paper.
3	7. Results: Demographics: Only demographics for group B provided in table 1?	A summary of Group A demographics is given in the main body of the text. More detailed participant information on Group A participants was not available, however, the authors will note this comment for any future large-scale trial.
3	8. Table 1: In the data column, percentages should be in brackets after the N. Eg. 21 (78%). The column header should specify what is contained in the column e.g. n (%). The only exception is the age of parent row, where you already specify what is being presented in the row subheadings, and mean should also have SD in brackets afterwards. E.g mean (SD) for the row subheading, and 31 (SD?) in the data column.	Thank you for highlighting this, we have amended Table 1 as per your suggestions.
3	9. Table 2: Same comments as for table 1, need column descriptor N (%), and effect estimate and CI should be presented in a separate column, with the column headed accordingly.	Thank you for highlighting this, we have amended Table 2 as per your suggestion. Due to table constraints, it was difficult to place CI's in a separate column, however we have amended the display to make this clearer.
3	10. Discussion: I would like to see more discussion around how your studies findings compared to other studies.	To expand on the study in the context of other published literature, we have added a paragraph to the 'Intervention Outcomes' section of the discussion.
3	11. Videos of Toothbrushing: You did not present findings on the toothbrushing videos, therefore this should not be discussed in your	We thank the reviewer for their comment, however, due to limitations with word count and scope, we have been unable to include the qualitative

	paper. You mention that this will be in another paper, therefore remove it from this one. Although, including the qualitative and video data could really make for a great paper? But if you decide not to include, then remove all mention of both parts entirely from this one.	and video data in one paper. As above (6), brushing duration taken from the videotapes was analysed as part of the quantitative analysis, therefore, the authors feel it is necessary to include this section. Parent-child interaction video will be addressed in a separate paper.
--	--	--

VERSION 2 – REVIEW

REVIEWER	Beckett , Deanna University of Otago, Oral Sciences
REVIEW RETURNED	25-Aug-2022

GENERAL COMMENTS	Thank you for revising the manuscript. I believe this is an indepth paper that would be of interest to the readers.
---